# Inter-Alpha Inhibitor Proteins Modify the Microvasculature after Exposure to Hypoxia–Ischemia and Hypoxia in Neonatal Rats

**DOI:** 10.3390/ijms24076743

**Published:** 2023-04-04

**Authors:** Francesco Girolamo, Yow-Pin Lim, Daniela Virgintino, Barbara S. Stonestreet, Xiaodi F. Chen

**Affiliations:** 1Department of Translational Biomedicines and Neuroscience (DiBraiN), University of Bari School of Medicine, 70124 Bari, Italy; 2ProThera Biologics, Inc., Providence, RI 02905, USA; 3Department of Pathology and Laboratory Medicine, Alpert Medical School of Brown University, Providence, RI 02905, USA; 4Women & Infants Hospital of Rhode Island, Alpert Medical School of Brown University, Providence, RI 02905, USA

**Keywords:** blood–brain barrier, hypoxia–ischemia, inter-alpha inhibitor proteins, laminin, tunneling nanotubes

## Abstract

Microvasculature develops during early brain development. Hypoxia–ischemia (HI) and hypoxia (H) predispose to brain injury in neonates. Inter-alpha inhibitor proteins (IAIPs) attenuate injury to the neonatal brain after exposure to HI. However, the effects of IAIPs on the brain microvasculature after exposure to HI have not been examined in neonates. Postnatal day-7 rats were exposed to sham treatment or right carotid artery ligation and 8% oxygen for 90 min. HI comprises hypoxia (H) and ischemia to the right hemisphere (HI-right) and hypoxia to the whole body, including the left hemisphere (H-left). Human IAIPs (hIAIPs, 30 mg/kg) or placebo were injected immediately, 24 and 48 h after HI/H. The brains were analyzed 72 h after HI/H to determine the effects of hIAIPs on the microvasculature by laminin immunohistochemistry and calculation of (1) the percentage area stained by laminin, (2) cumulative microvessel length, and (3) density of tunneling nanotubes (TNTs), which are sensitive indicators of the earliest phases of neo-vascularization/collateralization. hIAIPs mainly affected the percent of the laminin-stained area after HI/H, cumulative vessel length after H but not HI, and TNT density in females but not males. hIAIPs modify the effects of HI/H on the microvasculature after brain injury in neonatal rats and exhibit sex-related differential effects. Our findings suggest that treatment with hIAIPs after exposure to H and HI in neonatal rats affects the laminin content of the vessel basal lamina and angiogenic responses in a sex-related fashion.

## 1. Introduction

Perinatal hypoxic–ischemic and hypoxic brain injury are two of the most common causes of infant mortality and long-term neurologic disabilities [1,2,3,4], including intellectual disability and cerebral palsy, which place a large burden on society [4,5]. Hypothermia is the only approved therapy for hypoxic–ischemic encephalopathy (HIE) in full-term newborn infants. Unfortunately, this therapeutic approach is only partially protective [6,7,8], has a relatively narrow therapeutic time window of up to 6 h after birth, and can only be used to treat full-term infants [9,10,11,12,13]. Thus, there is an urgent need to develop additional novel therapies to reduce the prevalence and severity of HIE and the development of secondary complications in neonates.

Microvessels are key structures containing the morpho-functional components of the blood–brain barrier (BBB), which form a highly selective blood–brain vascular interface to maintain the homeostatic microenvironment of the central nervous system (CNS). Alterations in cellular (endothelial cells, astrocytes, pericytes, and microglia) and non-cellular (extracellular matrix molecules) components of the BBB-microvessels could contribute to hypoxia–ischemia (HI)- and hypoxia (H)-related brain injury and, consequently, represent an additional potential therapeutic target [4,14,15]. However, most of the research investigating the pathogenesis and treatment of perinatal brain injury after exposure to HI and H primarily focuses on excitotoxicity, oxidative stress, cell death, and depletion of energy stores in the first phase of injury (minutes to hours) and neuroinflammatory responses in the second phase of injury (hours to days) after the initial insult [4,14]. The responses of the developing microvasculature to HI- and H-related injury have received less consideration but could potentially affect outcomes and be modified by therapeutic strategies after exposure to HI and H in neonates [4]. In this regard, we have recently shown that neovascularization begins within 72 h after the exposure of fetal sheep to brain ischemia [16]. However, there is limited information regarding the effects of exposure to HI and H on neovascularization in the brain of neonatal rodents and human infants, with the exception of their effect on the retina [17,18].

Inter-alpha inhibitor proteins (IAIPs) are a family of structurally related anti-inflammatory, immunomodulatory proteins and have received increasing attention due to their contribution to many disease states, including HI-related brain injury in newborns and stroke in adult subjects [19,20,21]. In recent studies, we have shown that treatment with human plasma-derived IAIPs (hIAIPs) decreases neuronal cell death and neuroinflammation in male neonatal rats after exposure to HI [20,22] and attenuates lipopolysaccharide-related disruption of the BBB in adult male mice [23]. In addition, treatment with IAIPs has been shown to improve histopathological and behavioral outcomes, including decreasing the quantity of infarcted brain tissue in both male and female neonatal rats after exposure to HI brain injury [19,20,24,25,26,27]. However, information is not available regarding the effect of treatment with hIAIPs on brain microvasculature growth after exposure to HI- and H-related brain injury.

Based upon the above considerations, the objective of the current study was to investigate the effects of treatment with hIAIPs on the responses of the brain BBB-microvessels after exposure to HI- and H-related injury in neonatal rats. The Rice–Vannucci method was used to induce hypoxic and ischemic conditions in postnatal (P) 7 neonatal rats by exposure to right carotid artery ligation and 8% oxygen for 90 min. The brain of P7-10 rats is generally considered to be similar to the brain of near-term infants [28,29,30]. 

hIAIPs were injected intraperitoneally (30 mg/kg) at 0, 24, and 48 h after exposure to hypoxia and ischemia. The 30 mg/kg dose of hIAIPs was selected based on our previous studies showing that this dose ameliorated pathological brain injury, infarct volume, and neuroinflammation in neonatal rats 3 days after exposure to moderate HI [20,22]. Furthermore, we have shown that larger doses of hIAIPs (60 mg/kg and 90 mg/kg) did not provide additional benefits over the 30 mg/kg dose for reductions in infarct volumes or behavior tasks in neonatal rats after exposure to severe HI for 2 h [31]. The half-life of hIAIPs in the systemic circulation of neonatal rats is 12.4 h in sham-treated males, 9.6 h in sham-treated females, 23.1 h in HI-treated males, and 16.2 h in HI-treated female rats [32]. Therefore, we considered that a dosing interval of 0, 24, and 48 h after exposure to HI- and H-related injury represented a reasonable initial therapeutic trial. 

The brains were procured at P10. Laminin protein, a major component of the vessel basal lamina (VBL) molecular network, was immunolabelled to apply a laminin-based morphometric/laser confocal microscopy analysis to selected brain regions. The following parameters were analyzed: (1) the percentage of the area stained by laminin, (2) cumulative microvessel length (100 μm of vessel length/area/1 mm^2^), and (3) the density of tunneling nanotubes (TNTs). Measurements were analyzed in the caudate nucleus and putamen (corpus striatum), frontal, parietal, and occipital cerebral cortical lobes (neocortex), hippocampus, and hypothalamus (archi- and meso-cortex) in both the HI-right and H-left hemispheres. Laminin immunosignals facilitated morphometric analysis and quantification of the angiogenic responses to the HI and H exposure and also revealed the presence of TNT-like structures. TNTs are known to be ensheathed by a laminin-enriched basal membrane [33] and have been suggested to have a role in the earliest phases of neuroangiogenesis and neurovascular coupling as well [33,34].

## 2. Results

### 2.1. Effects of Treatment with hIAIPs on the Percent of Laminin-Stained Area in Brain Regions after Exposure of Neonatal Rats to Hypoxia–Ischemia and Systemic Hypoxia

Laminin is one of the key molecular components of vessel basal lamina in the CNS microvascular network [35]. Laminin immunohistochemical detection was used to visualize the entire blood vessel network and measure the percent of the laminin-stained area as a specific parameter of the vascular basal lamina in the brain [36]. Figure 1A contains the calculated values plotted as the laminin-positive pixel percent (%) of the total field area for males + females and male and female rats separately in the selected brain regions of the sham-treated, HI-/H-placebo (PL), and HI-/H-hIAIP treated (IAIP) experimental groups. 

Treatment with hIAIPs reduced the percent of laminin immunosignals in the systemic H-injured left caudate–putamen (Tukey’s HSD, *p* < 0.001) and HI-injured right caudate–putamen (Tukey’s HSD, *p* = 0.001) compared with the same regions in the males + females of H-PL and HI-PL groups, respectively. Treatment with hIAIPs reduced the percent of the laminin-stained area in the left caudate–putamen of the males (Tukey’s HSD, *p* = 0.030) but not in the right caudate–putamen (Tukey’s HSD, *p* = 0.123) compared with the H-PL and HI-PL groups, respectively. Treatment with hIAIP reduced the percent of the laminin-stained area in both left (Tukey’s HSD, *p* = 0.003) and right (Tukey’s HSD, *p* = 0.009) caudate–putamen of the females compared to the H-PL and HI-PL groups, respectively.

Treatment with hIAIPs reduced the percent of the laminin-stained area in the H-left (Tukey’s HSD, *p* = 0.013) and HI-injured right (Tukey’s HSD, *p* = 0.034) frontal cortices in the males + females but not in males compared with the H-PL and HI-PL groups, respectively, (Tukey’s HSD, all *p >* 0.05). Moreover, treatment with hIAIPs reduced the extent of the laminin-stained area in the H-left frontal cortex (Tukey’s HSD, *p* < 0.001) but not in the HI-right frontal cortex (Tukey’s HSD, *p* = 0.190) in the females. The percent of the laminin-stained area was lower in the hIAIP-treated group in the left parietal cortex of the males + females (Tukey’s HSD, *p* = 0.002) and in the left and right parietal cortices of the males (Tukey’s HSD, left: *p* = 0.038; right: *p* = 0.034), but not in the females (Tukey’s HSD, all *p >* 0.05) compared to the H-PL- and HI-PL-treated groups. Treatment with hIAIPs reduced the percent of the laminin-stained area in the H-left occipital cortex of the males + females (Tukey’s HSD, *p* = 0.033), whereas statistically significant changes were not observed in the HI-right occipital cortex (Tukey’s HSD, *p >* 0.05). Moreover, significant differences were not observed in either the male or female rats on the H-left or HI-right sides of the brain (all *p >* 0.05).

Treatment with hIAIPs reduced the percent of the laminin-stained area in the H-left and HI-right injured hippocampus of the male + female (left: Kruskal–Wallis, *p* = 0.001; right: Kruskal–Wallis, *p* = 0.008) and male (left: Tukey’s HSD, *p* = 0.017; right: Tukey’s HSD, *p* = 0.009) rats and the H-left (Kruskal–Wallis, *p* = 0.040) hippocampal regions of the female rat, compared to the H-PL- and HI-PL-treated groups, respectively. Similarly, treatment with hIAIPs reduced the percent of the laminin-stained area in both the H-left (Tukey’s HSD, *p* = 0.002) and HI-right (Tukey’s HSD, *p* = 0.016) hypothalamus in the males + females. However, the percent of laminin immunosignals was only reduced after treatment with IAIPs in the H-left hypothalamus of the males (Tukey’s HSD, *p* = 0.043) and females (Tukey’s HSD, *p* = 0.022), but not in the HI-injured right hypothalamus of the males or females (Tukey’s HSD, all *p >* 0.05). 

Regarding the interactive effects of the treatment x sex in the left or right brain, treatment x brain side in the males + females, males, and females, and treatment x brain side x sex, significant differences were not observed in the percent of the laminin-stained area observed in the caudate–putamen, frontal, parietal, or occipital cerebral cortices, hippocampus, or hypothalamus (factorial ANOVA, all *p >* 0.05). There were no differences in the interactive effects of brain side x brain region in the percent of the laminin-stained area in the sham group in either males or females factorial ANOVA, *p >* 0.05). Furthermore, differences were not observed in the interactive effects of treatment x brain side x sex in caudate–putamen, frontal, parietal, occipital cerebral cortices, hippocampus, or hypothalamus of the sham groups (factorial ANOVA, *p >* 0.05). 

Figure 1B shows the representative laminin staining of the H-left caudate nucleus in females. The H-related, increased laminin staining in the H-PL-treated caudate nucleus was reduced after hIAIP treatment. Additionally, laminin-positive TNT-like long filopodia were recognized as an aspect of re-activated angiogenesis in the H-PL-treated caudate nucleus (Figure 1B).

### 2.2. Treatment with hIAIPs Reduces Cumulative Blood Vessel Length in the Brains of Neonatal Rats after Exposure to Systemic Hypoxia

The effect of treatment with hIAIPs on the brain microvasculature was investigated further after exposure to HI- and H-injury by using laminin immunostaining to determine the cumulative microvessel length (expressed as multiples of 100 μm) per area (mm^2^) in the experimental groups (sham, H-PL/HI-PL, and H-IAIP/HI-IAIP and in the males + females, males, and females) and in the brain regions (caudate nucleus and putamen, frontal, parietal, and occipital cerebral cortices, hippocampus, and hypothalamus) (Figure 2A). Blood vessel cumulative length was significantly reduced after treatment with hIAIPs in the H-left caudate–putamen of the male + female (Tukey’s HSD, *p* = 0.005) and female (Tukey’s HSD, *p* < 0.001) rats and in the H-left parietal cortex of the male + female (Tukey’s HSD, *p* = 0.002) rats. Significant differences between study groups on the left and right sides in the male + female, male, and female rats were not detected in the other brain regions examined (Tukey’s HSD, all *p >* 0.05). Significant changes in cumulative blood vessel length were not observed in the right HI-exposed brain regions of the HI-PL compared with the sham-operated animals in the male + female and male or female groups.

The interactive effects of treatment x sex on the cumulative vessel length per image area were detected in left caudate–putamen (factorial ANOVA, F(2, 30) = 4.057, *p* = 0.028). However, interactive effects of the treatment x sex in the left or right brain, treatment x brain side in the males + females, males, and females, and treatment x brain side x sex, on the cumulative vessel length per area were not observed in any other brain region examined (factorial ANOVA, all *p >* 0.05). Differences were not observed in the interactive effects of brain side x brain region on the cumulative vessel length per image area in the sham group in either males or females (factorial ANOVA, *p >* 0.05). Furthermore, differences were also not observed in the interactive effects of treatment x brain side x sex in the caudate–putamen, frontal, parietal, occipital cerebral cortices, hippocampus, or hypothalamus (factorial ANOVA, *p >* 0.05). 

Figure 2B illustrates representative images of laminin-immunostained brain areas used for calculating the cumulative vessel length from the H-left caudate–putamen nuclei of sham, H-PL, and H-IAIP female animals. The cumulative vessel length per area was reduced after treatment with hIAIPs compared with those in the H-PL group. 

### 2.3. Treatment with hIAIPs Affects the Linear Density of Tunneling Nanotubes in More Brain Regions of the Female Than Male Neonatal Rats after Exposure to Systemic Hypoxia and Hypoxia–Ischemia

The observation of laminin-positive tunneling nanotube (TNT)-like structures bridging the gap between facing microvessels in H-PL/HI-PL rats prompted us to examine morphometrically and compare the linear density (cumulative length expressed in µm/acquired volume expressed as 10^6^ µm^3^) and the numerical density as determinants of potentially activated angiogenesis. In fact, pericyte-derived TNTs enclosed by a collagen IV-, fibronectin-, and laminin-enriched basal lamina [33,34] have been suggested to initiate vascular outgrowth, sustain neurovascular coupling, and contribute to cell-to-cell communication over long distances in normal and pathological conditions [37].

Figure 3A illustrates the results obtained in males + females, males, and females in the sham, H-PL/HI-PL, and H-IAIP/HI-IAIP groups of neonatal rats in the caudate–putamen nuclei, frontal, parietal, and occipital cortices, hippocampus, and hypothalamus. Rats treated with hIAIPs showed a reduced linear density of TNTs in the left and right caudate–putamen of the male + female (left: Kruskal–Wallis, *p* = 0.001; right: Kruskal–Wallis, *p* < 0.001) and female (left: Kruskal–Wallis, *p* = 0.004; right: Kruskal–Wallis, *p* = 0.005) rats compared with the H-PL/HI-PL groups. Treatment with hIAIPs also reduced the linear density of TNTs in the left frontal cortex of the male + female rats (Kruskal–Wallis, *p* = 0.017) and reduced the linear density of TNTs in the left and right parietal cortex of the male + female rats (left: Kruskal–Wallis, *p* = 0.013; right: Kruskal–Wallis, *p* = 0.003) and in the parietal cortex of the female rats (left: Kruskal–Wallis, *p* = 0.019; right: Kruskal–Wallis, *p* = 0.011). Treatment-associated reductions were observed in the hippocampus of the left and right sides of the male + female rats (left: Kruskal–Wallis, *p =* 0.043; right: Kruskal–Wallis, *p* = 0.021) rats. However, a significant reduction of the TNT linear density was only observed in the HI-right hippocampus of male rats (Kruskal–Wallis, *p* = 0.003).

Interactive effects of treatment x sex on the nanotube linear density were detected in the left parietal cortex (factorial ANOVA, F(2, 30) = 4.062, *p* = 0.027) and right hippocampus (factorial ANOVA, F(2, 30) = 5.937, *p* = 0.007). Interactive effects of the treatment x sex in the left or right brain, treatment x brain side in the males + females, males, and females, and treatment x brain side x sex on the nanotube linear density were not observed in the brain regions examined (factorial ANOVA, all *p >* 0.05). Furthermore, differences were not observed in the interactive effects of treatment x brain side x sex in the caudate–putamen, frontal, parietal, occipital cerebral cortices, hippocampus, or hypothalamus (factorial ANOVA, *p >* 0.05). 

Figure 3B demonstrates representative images of laminin-immunostained TNTs from the right side of caudate–putamen nuclei of the sham, HI-PL, and HI-IAIP female rats. These aspects of re-activated angiogenesis were observed mainly in the HI-PL group and appeared reduced after treatment with hIAIPs.

### 2.4. Treatment with hIAIPs Affects Tunneling Nanotube Numeric Density in More Brain Regions of the Female Compared with the Male Neonatal Rats after Exposure to Systemic Hypoxia and Hypoxia–Ischemia

Figure 4A contains the numerical density (number of identified TNT-like structures/acquired volume 10^6^ µm^3^) of TNTs for the males + females, males, and females in the sham, H-PL/HI-PL, and H-IAIP/HI-IAIP groups in the caudate–putamen nuclei, frontal, parietal, and occipital cortices, hippocampus, and hypothalamus. Treatment with hIAIPs reduced the TNT numerical density in the H-left and HI-right caudate–putamen of the male + female (left: Kruskal–Wallis, *p* = 0.006; right: Kruskal–Wallis, *p* < 0.001) and female (left: Kruskal–Wallis, *p* = 0.007; right: Kruskal–Wallis, *p* = 0.005) rats compared with the H-PL/HI-PL groups. Reduced values were also observed in the H-left and HI-right parietal cortex of the male + female (left: Kruskal–Wallis, *p* = 0.002; right: Kruskal–Wallis, *p* = 0.002) and in the female (left: Kruskal–Wallis, *p* = 0.046; right: Kruskal–Wallis, *p* = 0.010) rats, and in the H-left and HI-right hippocampus of the male + female (left: Kruskal–Wallis, *p =* 0.011; right: Kruskal–Wallis, *p* = 0.031) rats. The numeric density of the TNTs was reduced in the HI-right hippocampus after treatment with hIAIP (Kruskal–Wallis, *p* = 0.005).

The interactive effects of treatment x sex on the nanotube numerical density were detected in the left (factorial ANOVA, F(2, 30) = 5.756, *p* = 0.008) and right hippocampus (factorial ANOVA, F(2, 30) = 5.394, *p* = 0.001). The interactive effects of the treatment x brain side were detected in the hippocampus of the male + female (factorial ANOVA, F(2, 60) = 5.106, *p* = 0.009) and female (factorial ANOVA, F(2, 30) = 10.504, *p* < 0.001) rats. Furthermore, the interactive effects of the treatment x brain side x sex on the nanotube numeric density were found in the hippocampus (factorial ANOVA, F(2, 60) = 9.495, *p* < 0.001) and between caudate–putamen, frontal, parietal, occipital cerebral cortices, hippocampus, and hypothalamus (factorial ANOVA, F(10, 360) = 1.917, *p* = 0.042). 

Figure 4B demonstrates representative images of laminin immunostained TNTs from fields used to measure TNT numeric density in the right caudate–putamen nuclei of sham, HI-PL, and HI-IAIP female animals. TNTs (arrowheads) were reduced after treatment with hIAIPs compared to that in the HI-PL group. 

The results are schematically summarized in Table 1. The analysis of the chosen morphometric parameters (laminin immunostained area percentage, cumulative vessel length, TNT linear density, and TNT numerical density) was performed on selected brain regions of rats on postnatal day 10. The arrows compare the values obtained in hIAIP-treated versus the PL-treated rats. Treatment with hIAIPs reduced the percent of laminin immunostained areas after exposure to systemic H-left and HI-right injury, reduced the cumulative vessel length in a few brain regions after exposure to systemic H (left hemisphere), but not HI (right hemisphere), and diffusely reduced TNT linear density and numeric density. These morphometric parameters were selected to examine possible angiogenic responses. Interestingly, the reduced values were detected in relatively more brain regions in the female compared with the male neonatal rats after systemic hypoxic (left hemisphere) and hypoxic–ischemic (right hemisphere) injury. Taken together, the results suggest that treatment with hIAIPs after exposure to H and HI in neonatal rats has effects on the laminin content of the VBL as well as on the angiogenic responses in a phylogenetically- and sex-related fashion [25].

## 3. Discussion

In the current study, we found that systemic treatment with blood-derived hIAIPs after exposure to systemic hypoxia- and/or hypoxia–ischemia-related brain injury affects the laminin area percent of the vessel basal lamina, cumulative microvessel length, and linear and numerical densities of TNTs and also exhibited some sex-related differential effects in neonatal rats. 

The well-established Rice–Vannucci rat model was used, in which the hypoxic component of the insult is systemic and affects the entire body along with both the left and right hemispheres, and the ischemic component is confined primarily to the right hemispheric injury. Laminin is a non-collagenous component of the vessel basal lamina, and its expression is closely related to angiogenesis along with the functional and structural maturation of microvessels during brain development [4,38,39,40]. Laminin has been extensively used as an important marker of microvasculature in rodents [41,42] and other species, including humans [41,43,44]. It is a valuable determinant of the microvasculature because it is able to reveal not only endothelium-associated morphology of the microvessels but also BBB/angiogenic morpho-functional features of the microvessels [45,46,47]. Laminin is also a molecular component of the neurovascular unit (NVU) that is functionally associated with CNS microvessel performance in both physiological and pathological conditions [45,48]. Therefore, we examined the laminin content of the vessel basal lamina quantitatively to reveal potential structural changes in the microvessel wall after exposure to hypoxic and hypoxic–ischemic injury. Laminin immunostaining was also used to calculate the extension of the microvascular network through analyses of the cumulative vessel length. In addition, laminin immunostaining also facilitated the determination of the number and density of the TNT structures. 

Significantly fewer laminin-stained positive areas were observed in hIAIP-treated than in PL-treated hypoxic and/or hypoxic–ischemic brain regions. In addition, we observed less cumulative microvessel length in hIAIP-treated male + female and female caudate putamen and male + female parietal cortex, which were observed more frequently in the hIAIP treated hypoxic than in the hypoxic–ischemic brain regions (Table 1). These findings suggest that the structural integrity of the vessel basal lamina was affected by treatment with hIAIPs in the neonatal brain after exposure to hypoxia- and/or hypoxia–ischemia-related brain injury. These findings could be further investigated by determining pericyte-specific laminin isoforms [49,50].

Consistent with our findings in the brain microvessels after hypoxia and hypoxia–ischemia, these potent post-ischemic effects of hIAIPs on the basal lamina of cerebral vessels were also observed in pulmonary endothelial cells, in which hIAIP treatment ameliorated endothelial inflammation in a mouse model of sepsis [51]. Although we cannot determine the precise mechanism(s) by which treatment with hIAIPs potentially inhibited the angiogenesis after exposure to hypoxia (left brain) and hypoxia–ischemia (right brain), IAIPs have been shown to bind to the high molecular weight hyaluronic acid [52,53,54], which has been shown to have anti-angiogenic activity [55,56,57]. Nonetheless, we cannot determine whether the responses of the microvasculature to treatment with hIAIPs contributed to the significant neuroprotection that we have previously reported [20,22,24,25]. This is in part because it has not been completely clarified whether the hypoxia-induced angiogenesis and/or hypoxia–ischemia-induced increases in the density of proliferating brain microvessels have protective or detrimental effects on the neonatal brain after exposure to injury. Moreover, it is difficult to ascertain from the current study whether there is less injury to the microvasculature because of treatment with hIAIPs per se or whether there is less parenchymal injury and, consequently, less injury to the microvasculature [20,22,24,25].

Hypoxia is a key regulator of angiogenesis, and hypoxia-induced angiogenesis is thought to be protective because it increases cerebral blood flow and delivers more oxygen and nutrients to oxygen-deprived tissues [4]. Moreover, the BBB microvasculature consists of specialized endothelial cells surrounded by astrocytes and pericytes and has a crucial role in brain homeostasis. Previous work has shown that pericytes enhance cellular crosstalk within the neurovascular unit, are crucial in preserving barrier integrity under hypoxic conditions, and play an important role during severe and prolonged hypoxia [58]. In addition, blood containing toxic metabolites produced by the hemisphere ipsilateral to the ligated carotid artery (right hemisphere) could also affect the contralateral hemisphere (left hemisphere) via the circle of Willis in the Rice–Vannucci model. These factors potentially might underlie our findings, suggesting that there was a relatively greater response in the hypoxic left side compared with the hypoxic–ischemic right side of the brain to form TNTs potentially to stimulate vessel neoformation and vessel collateralization (Figure 4 and Table 1) [34]. Nonetheless, hypoxia can also be associated with adverse effects, such as alterations in the BBB and the function of the neurovascular unit (NVU) that are not entirely beneficial [4,45]. Although we have previously identified increases in claudin-5 expression 6 h and 48 h after exposure to hypoxia in the hippocampus of neonatal rats [45], we have not examined the expression of tight junction proteins 72 h after exposure to hypoxia or hypoxia–ischemia or the potential effects of treatment with hIAIPs in the current study. In addition, hypoxia-induced angiogenesis has been suggested to have potentially adverse effects on brain injury in neonates after severe fetal hypoxic exposure because the newly formed blood vessels are fragile, prone to rupture, and bleeding, potentially resulting in hemorrhage [4,59]. 

hIAIPs-induced changes in cumulative microvessel length were only observed in the brain regions exposed to hypoxia. One of the potential explanations is that we examined changes in cumulative vessel length at 72 h after exposure to the injury. Previous studies have shown that proliferating brain microvessels increased as early as three days after hypoxia–ischemia [16,60,61], and angiogenesis-promoted neovascularization occurred within several days after ischemic brain injury in neonatal rodents [62]. We cannot rule out the possibility that vascular growth could have occurred before and/or continued beyond 72 h after exposure to hypoxia–ischemia. Nonetheless, our results can also be interpreted to suggest that brain microvessels are in a dynamic balance between growth proliferation and inhibition to maintain the function of the microvascular structure after injurious events.

Hypoxia–ischemia-related injury is more severe than hypoxia alone. In addition to resulting in energy failure, hypoxia–ischemia results in the accumulation of toxic metabolites, such as lactic acid, that are normally removed by the circulation. Excessive tissue lactic acidosis can result in irreversible cellular damage, including in endothelial cells [63]. Furthermore, the results of microvessel responses to hypoxia–ischemia and hypoxia-related injury vary after different hypoxia/hypoxia–ischemia reoxygenation and reperfusion periods, brain regions, and animal models. Decreases in vessel density were found in the caudate nucleus of fetal sheep after exposure to umbilical cord occlusion [14,64] and the cerebral cortex of rodents [65] and newborn piglets [66] after exposure to hypoxia–ischemia-related injury. In contrast, changes were not observed in blood vessel density and morphology in the cerebral cortex of fetal sheep after exposure to umbilical cord occlusion [64]. However, increased neovascularization and microvessel density were observed in the fetal ovine cerebral cortex within 72 h of ischemia [16] and in the cerebral cortex and hippocampus of neonatal rats 48 h after exposure to severe HI-related injury induced by permanent common carotid artery ligation plus 2 h of hypoxic injury [45]. The reason for these discrepancies could also be related to different rates of neoangiogenesis among different species during pre- and postnatal development.

TNTs represent novel structures that facilitate communication between distant cells. They can be formed by many cell types, including epithelial, neuronal, and almost all immune cells [67]. TNTs exert detrimental roles in cell differentiation and immune function via intercellular exchanges of signals, molecules, organelles, aggregated misfolded proteins [68,69,70,71], and pathogens [67,72,73,74,75]. TNT-like structures have long been observed to augment inflammatory conditions and facilitate intercellular propagation of toxic agents as well as other molecules during stress and pathological conditions [72,76]. Furthermore, pericyte-derived TNTs are involved in normal and tumoral neuroangiogenesis [34] and cell-to-cell signaling between the cells of the neurovascular unit [33]. Healthy mitochondria can also be delivered to injured astrocytes via TNTs. This process could be an important mechanism for the protection and repair of injured brain tissue during the perinatal period [77,78]. In the current study, we demonstrated that systemic treatment with hIAIPs reduces brain linear and numeric densities of TNTs in neonatal rats after exposure to hypoxia and hypoxia–ischemia-related injury. 

Remarkably, the effects of hIAIPs on the densities of TNTs were mainly observed in female animals, suggesting that the responses of TNT formation to treatment with hIAIPs in neonatal hypoxia–ischemia brain injury could be sex-dependent. We have previously shown that hIAIPs exert neuroprotective effects on neonatal rats exposed to hypoxia–ischemia-related brain injury in a sex-dependent manner [20,24]. Although sex differences have been reported in vascular and BBB-related changes under different pathological conditions in both rodent and human tissues [79,80,81], whether the sex-related changes in TNT formation in response to treatment with hIAIPs correlate with the sex differences in neuroprotection afforded by hIAIPs after HI brain injury cannot be discerned by the current study. However, it has been shown that there are early sex differences in the immune-inflammatory responses to neonatal HI brain injury [82]. Immune-inflammatory responses could play a potential role in the sex-dependent TNT formation responses to hIAIP treatment because IAIPs represent a family of structurally related anti-inflammatory, immunomodulatory proteins. This could be a potential mechanism underlying the changes after HI brain injury in neonatal rats. However, elucidation of the mechanism(s) underlying the sex-dependent changes in TNT formation after hypoxia–ischemia requires further investigation. 

Disruption of the BBB is a critical component of ischemia-related brain damage in the fetus [83]. Treatment with hIAIPs could potentially decrease the concentration of molecules in the systemic circulation known to disrupt the integrity of the BBB [14,84]. We have previously shown that intraperitoneal administration of hIAIPs protects against LPS-induced disruption of the BBB in adult male mice [23] and may preserve BBB function through mechanisms associated with systemic cytokine suppression during inflammatory conditions [84]. Numerous papers suggest that IAIPs can suppress inflammation and cytokine production in a variety of animal models [85,86]. In addition, we have previously shown that anti-cytokine antibodies against IL-1β and IL-6 attenuate ischemia-related disruption of the BBB in fetal sheep [87,88,89]. Furthermore, IAIPs bind hyaluronic acid, which also has been shown to affect BBB function [90]. Therefore, it remains likely that IAIPs could also potentially attenuate H/HI-related disruption of the BBB in neonatal rats [91]. Consequently, the impact of hIAIPs on the structures, such as tight junction proteins and the function of the BBB after exposure to H/HI-related brain injury, remains an important area for future study [45]. 

There are several limitations to our study and opportunities for further investigation. It would be of great interest to perform high-resolution micro-CT images of the brain vasculature because laminin is a component of the basal lamina and does not label the intraluminal space, which makes quantification of vessel volume and density by immunohistological methods challenging [92,93,94,95]. Although a covalent complex of hyaluronan and the heavy chain of IAIPs has been demonstrated to have anti-angiogenic activity in vitro [56], we did not examine the key molecules for regression and re-growth of brain microvasculature such as vascular endothelial growth factor (VEGF), placental growth factor, and acidic-fibroblast growth factor because of the sample limitations in the current study. Therefore, the mechanism(s) underlying the hIAIP-related inhibition of the post-ischemic angiogenesis require(s) additional investigation, including studying key aspects of hIAIPs on angiogenic phenotypes using brain endothelial cells and further investigating the internal endothelial and external parenchymal basal laminae within the microvessels after exposure to hypoxia and hypoxia–ischemia along with the potential effects of neuroprotective of agents such as hIAIPs [96]. 

Although we have identified important microvascular changes after exposure to hypoxia and hypoxia–ischemia and shown that hIAIPs attenuate some of these changes, the molecular basis for these modifications awaits further investigation. However, we speculate that some of the mechanism(s) potentially involved could be related to the early expression of growth factors after hypoxia–ischemia. It has been shown that early VEGF expression after hypoxic insults does not necessarily have beneficial neuroprotective effects but could have adverse effects on the BBB, including increased permeability to peripheral immune cells that can gain access to the developing brain and arrested oligodendrocyte differentiation [97,98]. In addition, hyaluronic acid binds to IAIPs and has the capacity to selectively bind the angiogenic isoform of VEGF [99] so that IAIPs could sequester this angiogenic isoform of VEGF in the early phases of hypoxic–ischemic insults. Nonetheless, additional investigation is required to determine the molecular mechanism(s) underlying the microvascular changes that we have observed. 

In conclusion, systemic treatment with hIAIPs affected the laminin-positive vessel basal lamina, including the laminin area percentage, cumulative microvessel length, and linear and numerical densities of TNTs in neonatal rat brain after exposure to hypoxia- and/or hypoxia–ischemia-related brain injury and may exhibit sex-related differential effects. Our findings suggest that treatment with hIAIPs after exposure to hypoxia and hypoxia–ischemia hIAIPs can attenuate the hypoxia and hypoxia–ischemia-related remodeling/thickening of the vessel basal lamina in neonatal rats. This effect is evidenced by the reductions to values that are similar to the laminin content values under normoxic conditions in control neonatal rats [100]. 

In addition, sex differences in the response of the brain to hypoxic and hypoxic–ischemic treatment may reflect a somewhat different timetable in male and female brain microvascular development [101], which appears to be evident in the responses to the hypoxic and ischemic conditions of the female neocortex versus the male archi-/paleo-cortex, according to the biogenetic premise “ontogeny recapitulates phylogeny” (Ernst Haeckel, 1866). Although very little is known about sex differences in BBB-microvessel adaptations to hypoxic and hypoxic–ischemic injury, increased attention to sex differences may elucidate potential novel biological targets and differential therapeutic approaches. 

## 4. Materials and Methods

### 4.1. Production and Purification of Human Inter-Alpha Inhibitor Proteins (hIAIPs)

The hIAIPs were extracted from pooled fresh frozen human plasma (Rhode Island Blood Center, RI, USA) or other commercial sources. The procedures of hIAIP purification and concentration have been described in detail in previous publications [20,32,102]. A 50–65% yield from 1 L of plasma produces ca. 100 mg of highly pure (>90%) hIAIPs. The purity was confirmed by using SDS-PAGE, Western immunoblot, and competitive immunoassay [102,103]. The biological activity was measured by the ability of hIAIPs to inhibit the hydrolysis of the substrate N-Benzoyl-L-arginine-p-nitroaniline HCl (MilliporeSigma, St. Louis, MO, USA) by trypsin [104]. Moreover, the endotoxin level in the purified product was monitored using a limulus amebocyte lysate endotoxin-based chromogenic test (Pierce Biotechnology, ThermoFisher Scientific, Waltham, MA, USA) [32,102,104]. One individual lot of purified hIAIPs was used for the entire study to avoid potential variations in the purity and quantity of the hIAIPs injected.

### 4.2. Experimental Animals and Hypoxic–Ischemic (HI) Procedures

The present study was performed after approval from the Institutional Animal Care and Use Committees of the Alpert Medical School of Brown University and Women & Infants Hospital of Rhode Island. All experimental procedures were carried out following the National Institutes of Health Guidelines for the Use of Experimental Animals.

Wild-type Wistar rats were obtained from Charles River Laboratories (Wilmington, MA, USA) on embryonic pregnancy day 15 or 16. Pregnant rats were housed in a temperature-controlled, 12 h light/dark-cycled facility with ad libitum access to food and water in the Care Facility at Brown University. The dates upon which the pups were delivered were designated as P0. On P1, litters were randomly culled to ten and balanced such that there were approximately equal numbers of males and females to reduce inter-litter variability. On P7, the rats were randomly assigned to one of the following groups: sham-operated controls (sham), placebo (PL)-, and hIAIPs-treated pups with exposure to right carotid artery ligation and hypoxia (left brain: H-PL and H-IAIP; right brain: HI-PL and HI-IAIP). The right carotid artery ligation and hypoxia were performed using the Rice–Vannucci method as previously described [20,105,106]. Briefly, the P7 pups were anesthetized with isoflurane (induction: 4%; maintenance: 2%) in oxygen and underwent right common carotid artery ligation, whereas the sham pups received a neck incision only [105,107]. After recovery from surgery, the pups were returned to their dams for 1.5–3 h. Then, the pups exposed to right carotid artery ligation were placed in a temperature-controlled airtight chamber (Biospherix, Parish, NY, USA) with 8% humidified oxygen and balanced nitrogen for 90 min. One non-ligated sentinel pup per litter had a rectal temperature probe (RET-4, Physitemp, Clifton, NJ, USA) placed to monitor body temperature every 10 min during the hypoxic exposure. The rectal temperature was maintained at close to 36.0 °C [8,32,108]. The sentinel pup was not included in further investigations because the stress of the rectal probe placement can alter the outcome of the HI exposure [108,109]. The rectal temperature has been shown to accurately reflect brain temperature in the HI rodent model [110,111]. The sham pups were placed in a similar container and remained in room air for 90 min. 

After recovery from surgery and hypoxia, the pups were given three intraperitoneal (I.P.) injections of 30 mg/kg of hIAIPs or equivalent volumes of placebo (phosphate-buffered saline, PBS) immediately (time 0), 24, and 48 h after termination of hypoxia (Figure 5A). This treatment paradigm was selected based on our previous studies showing that the same dose and injection period of hIAIPs protected neonatal rat brains from HI-related insults [20,22]. The pups were sedated with a cocktail of ketamine (74 mg/kg, I.P.) and xylazine (4 mg/kg, I.P.) at 72 h after HI. The brains were perfused with ice-cold PBS and 4% paraformaldehyde (PFA, Electron Microscopy Sciences, Hatfield, PA, USA) via cardiac puncture at a flow rate of 3 ml/min. Thereafter, the brains were removed and post-fixed with PFA for 24 h and stored in 30% sucrose in phosphate buffer (0.1 M, MilliporeSigma, St. Louis, MO, USA) at 4 °C before paraffin embedding. The paraffin-embedded brain tissues were sectioned in coronal planes at 10 µm thickness. Three coronal sections per brain (frontal section: bregma 1.28 ± 1.16 mm; middle section: bregma −3.36 ± 1.40 mm; posterior section: bregma −6.00 ± 0.72 mm) were utilized to standardize the immunohistochemical analysis (Figure 5B) [112,113]. All neonatal rats survived the experimental protocols and were included in the analysis. 

In our previous publication [20], we used Nissl cresyl violet staining to identify neuronal damage in neonatal rats using the identical hypoxic–ischemic and hIAIP treatment procedures as in the current study. One Nissl-stained image, which is distinct from our previous publication [20], is shown in Appendix A in order to illustrate the extent of the HI-related damage and neuroprotection provided by hIAIPs. 

### 4.3. Histology and Immunofluorescence

The coronal brain sections from the sham, H-/HI-PL, and H-/HI-IAIP groups were processed for immunohistochemical, high-resolution confocal immunofluorescence and morphometric analyses. The initial analysis was carried out by immuno-enzymatic method for the acquisition of magnification and wide optical fields to reveal the vascular basal lamina and to examine the percent of the laminin-stained area and cumulative vessel length (CVL). Single immunostaining was carried out with rabbit anti-laminin antibody (L9393, MilliporeSigma, St. Louis, MO, USA). Briefly, sections were rehydrated and processed for heat-mediated antigen retrieval (Dako antigen retrieval buffer, Carpinteria, CA, USA) at 98 °C for 60 min. The sections were then sequentially incubated with the following steps: (1) 1% H_2_O_2_ (MilliporeSigma, St. Louis, MO, USA); (2) blocking buffer (BB; PBS, 1% bovine serum albumin, and 2% fetal calf serum; Dako Italia, Milan, Italy) with 10% normal goat serum (Dako) and 0.1% Triton X-100 (Merck, Darmstadt, Germany) for 30 min at room temperature (RT); (3) rabbit anti-laminin antibody (1:200 in BB) overnight at 4 °C; (4) biotinylated goat anti-rabbit secondary antibody (1:400 in BB, Vector Laboratories, Burlingame, CA, USA) for 50 min at RT; (5) HRP-streptavidin (1 µg/mL; Vector Laboratories) for 45 min at RT and 3-amino-9-ethylcarbazole (AEC) substrate (Vector Laboratories) for 25 min at 37 °C. The sections were washed with PBS 3 times for 5 min each. Then the sections were coverslipped with Glycergel (Dako). 

After the acquisition of multiple fields with a digital camera (SPOT Insight Color, Diagnostic Instruments, Sterling Heights, MI, USA) on an Olympus Vanox T microscope (Olympus, Milan, Italy), using a 10× lens, the same sections were immunolabeled for high-resolution confocal analysis to disclose TNTs. Briefly, sections were rehydrated, uncover-slipped, and processed for antibody elution by heating the slides in a buffer (pH 2) containing 25 mM glycine-HCl and 1% SDS on a platform shaker in an incubator at 50 °C for 60 min. The sections were then sequentially incubated with the following steps: (1) 0.5% Tween-20 in PBS 3 times for 5 min each at RT; (2) 1% sodium tetrahydridoborate (NaBH_4_) for 2 min at RT; (3) BB for 30 min at RT; (4) primary antibody solution of rabbit anti-laminin antibody diluted 1:200 in BB plus 10% normal goat serum (Dako) and 0.1% Triton X-100 overnight at 4 °C; (5) biotinylated goat anti-rabbit secondary antibody (1:400 in BB, Vector Laboratories) (6) Alexa 555 fluorophore-conjugated goat anti-rat (1:400 in BB, Invitrogen) for 50 min at RT; (7) Alexa 488 fluorophore-conjugated streptavidin (1:300 in BB, Invitrogen) for 45 min at RT. The sections were post-fixed in 4% PFA for 10 min after immunolabeling. The sections were washed with PBS 3 times for 5 min after each incubation step. Finally, the sections were coverslipped with Vectashield mounting medium (Vector Laboratories) and sealed with nail varnish. Negative controls were prepared by omitting the primary antibodies or mismatching the secondary antibodies. Double immunolabeled sections were examined under the Leica TCS SP5 confocal laser-scanning microscope (Leica Microsystems, Mannheim, Germany) using a sequential scan procedure. Confocal images were taken at 500 nm intervals through the z-axis of the sections with a 63× oil lens. Z-stacks of serial optical planes (projection images) and single optical planes were analyzed by Leica confocal software (Multicolor Package; Leica Microsystems) and Image J (National Institutes of Health, Bethesda, MD, USA).

### 4.4. Morphometry and Quantitative Analysis

Analyses included quantification as follows: (1) the percent of the laminin-stained area, (2) the cumulative laminin-positive vessel length (expressed as multiples of 100 μm of the vessel length)/area (1 mm^2^), (3) the cumulative laminin-positive nanotube linear density in µm/acquired volume expressed as 10^6^ µm^3^), and (4) the nanotube numerical density (number of identified nanotubes/acquired volume). All quantifications were performed by an observer who was not aware of the group designations, using computer-aided morphometry on the microscopic images by the analysis functions of ImageJ. An average of 12 fields per section from each hemisphere (n = 3 sections per brain) were acquired from the caudate and putamen nuclei, frontal, parietal, and occipital cortices, hippocampus, and hypothalamus. Images of the selected immunoreactive areas were acquired using a 10× lens (field total area measuring 975,440 µm^2^ at 0.71 µm per pixel) for each laminin immunolabeled bright field. Confocal images acquired using 40× and 63× oil lenses (area 150,156 µm^2^ and 60,516 µm^2^ at 0.24 µm per pixel) were measured on laminin single-channel and z-stacks of the 18 single optical planes. Appendix A illustrates our analysis of the microvessel measurements in detail.

### 4.5. Statistics

All results were expressed as mean ± standard deviation (SD). The Shapiro–Wilk’s W normality test was first applied to test the distributions of all data obtained from the experiments. One-way ANOVA was used to detect differences in each brain region between groups in normally distributed data. The interactive effects of multiple categorical independent variables such as (1) the effects of treatment x brain side within each males + females, males, and females, (2) the effects of treatment x sex within each left and right brain side, and (3) the effects of treatment x brain side x sex were analyzed with factorial ANOVA. When a significant difference was detected, Tukey’s honest significant difference (HSD) test was used to identify group differences. If the data were not normally distributed, the data were first log_10_-transformed to satisfy normality (Shapiro–Wilk normality test) and homogeneity of variance (Levene’s test) assumptions for ANOVA. Otherwise, the non-normally distributed data and the data that cannot be log_10_-transformed were analyzed by Kruskal–Wallis ANOVA and median test followed by Dunn’s test for comparing multiple independent groups. All statistical analyses were performed using the STATISTICA package (TIBCO Software Inc., Palo Alto, CA, USA), and a *p* < 0.05 was considered to indicate statistical significance.

## Figures and Tables

**Figure 1 ijms-24-06743-f001:**
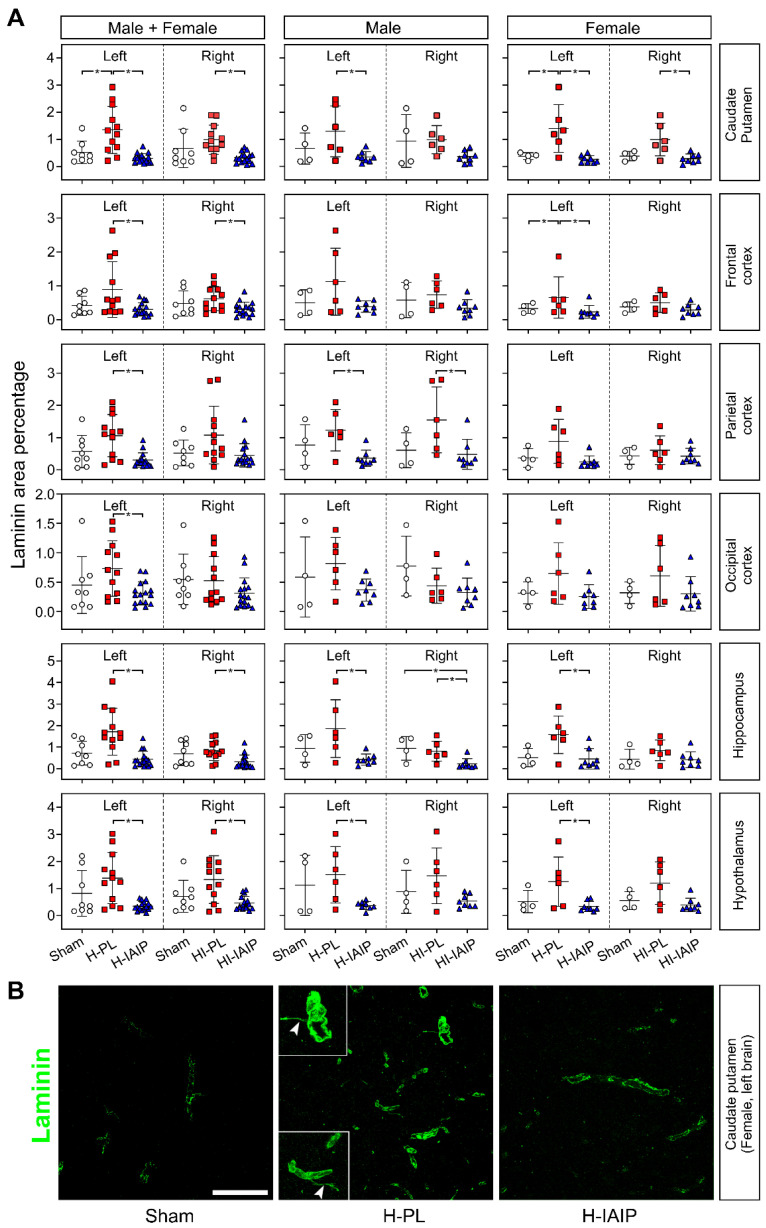
The effects of systemic treatment with hIAIPs on the percent of the laminin-stained area in neonatal rats after exposure to systemic hypoxia (H)-left and hypoxia–ischemia (HI)-right sides of the brain. (**A**) Quantification of the percent of the laminin-stained area for the total image of the laminin-stained area in the experimental groups in the left and right hemispheres in the total group of males plus females and separately for the male and female pups. Data are mean ± SD. * *p* ≤ 0.05 for indicated comparisons. Male: sham = 4, HI-PL/H-PL = 6, HI-IAIP/H-IAIP = 8. Female: sham = 4, HI-PL/H-PL = 6, HI-IAIP/H-IAIP = 8. (**B**) Representative confocal microscopic images of caudate/putamen nuclei in the systemic hypoxic left hemisphere of the females immunostained for laminin. Compared with placebo-treated systemic hypoxia animals (H-PL), treatment with hIAIPs (H-IAIP) reduced the increased laminin immunoreactivity to the level observed in sham-operated animals (sham). Laminin-positive TNT-like structures are recognizable as thin, long filopodia (arrowheads in insets). Scale bar: 100 µm.

**Figure 2 ijms-24-06743-f002:**
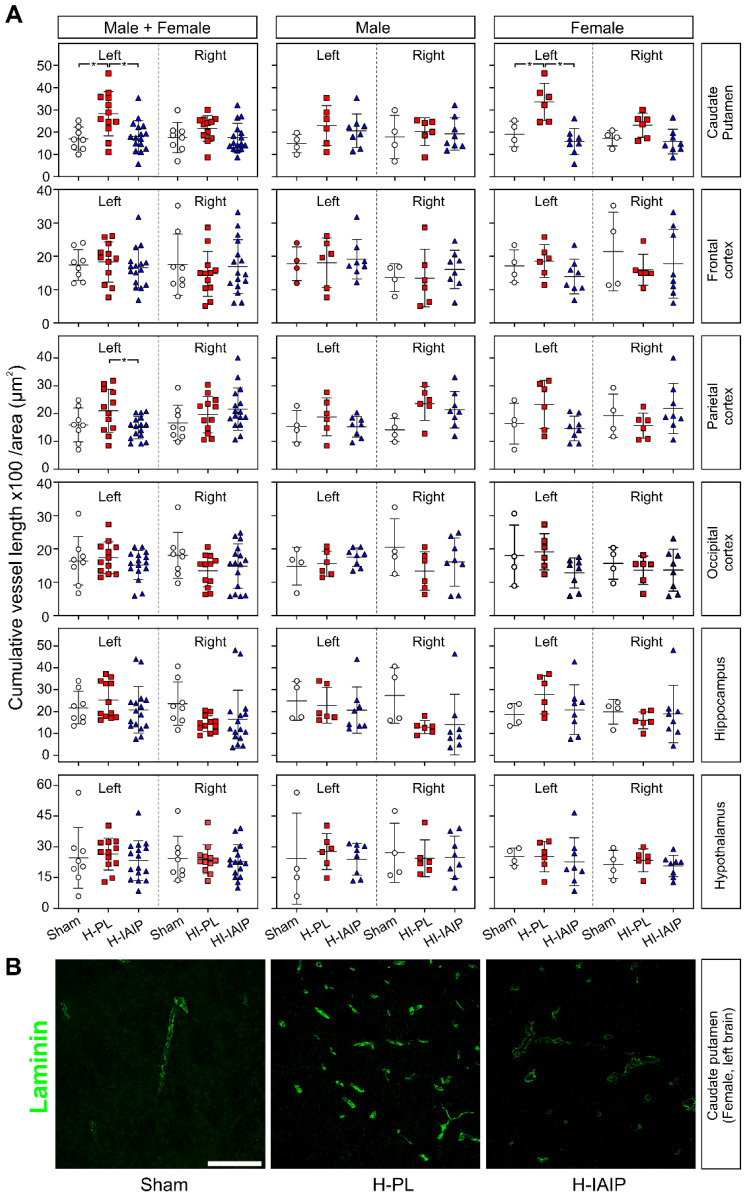
The effects of systemic treatment with hIAIPs on cumulative vessel length in neonatal rats exposed to systemic hypoxia (H, left side of the brain) and hypoxia–ischemia (HI, right side of the brain). (**A**) Quantification of the vessel length parameters in the total group of males + females and separately for the male and female pups. Data are mean ± SD. Male: sham = 4, H-PL/HI-PL = 6, H-IAIP/HI-IAIP = 8. Female: sham = 4, H-PL/HI-PL = 6, H-IAIP/HI-IAIP = 8. * *p* ≤ 0.05 for indicated comparisons. (**B**) Representative confocal microscopy images of H-left caudate-putamen nuclei immunostained for laminin in the female rats. Treatment with hIAIPs (H-IAIP) reduced the increased cumulative vessel length observed after exposure to hypoxia to the amounts observed in sham-operated animals (sham). Scale bar: 100 µm.

**Figure 3 ijms-24-06743-f003:**
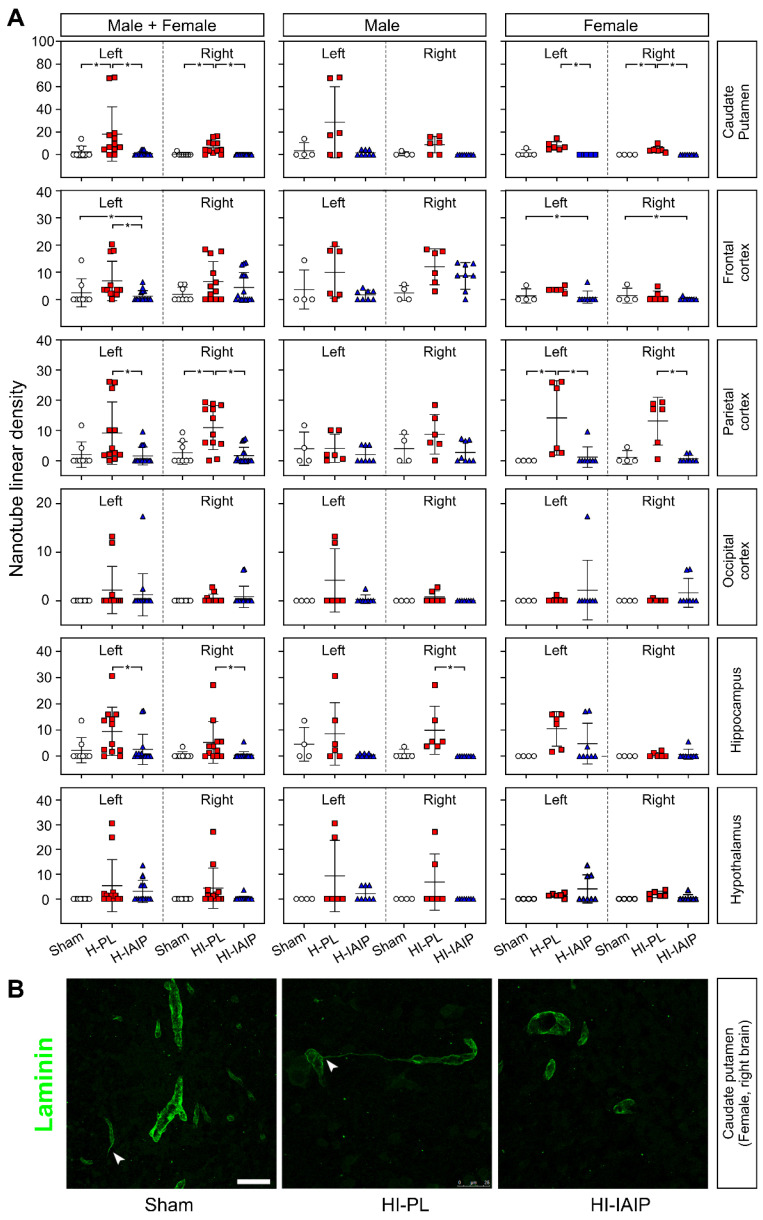
The effects of systemic treatment with IAIPs on TNT linear density in neonatal rats after exposure to systemic hypoxia and hypoxia–ischemia. (**A**) Quantification of cumulative TNT linear density in the experimental groups in the selected regions of the left and right hemispheres and total group of males + females and separately for the males and females. Data are mean ± SD. Male: sham = 4, H-PL/HI-PL = 6, H-IAIP/HI-IAIP = 8. Female: sham = 4, H-PL/HI-PL = 6, H-IAIP/HI-IAIP = 8. * *p* ≤ 0.05 for indicated comparisons. (**B**) Representative confocal microscopic images of laminin immunostained HI-right caudate–putamen in the female neonatal rats, showing the presence of TNTs connecting two microvessels (arrowhead, HI-PL). In sham-treated and HI-IAIP-treated brains, these angiogenic features are rarely detected (arrowhead, sham). Scale bar: 25 µm.

**Figure 4 ijms-24-06743-f004:**
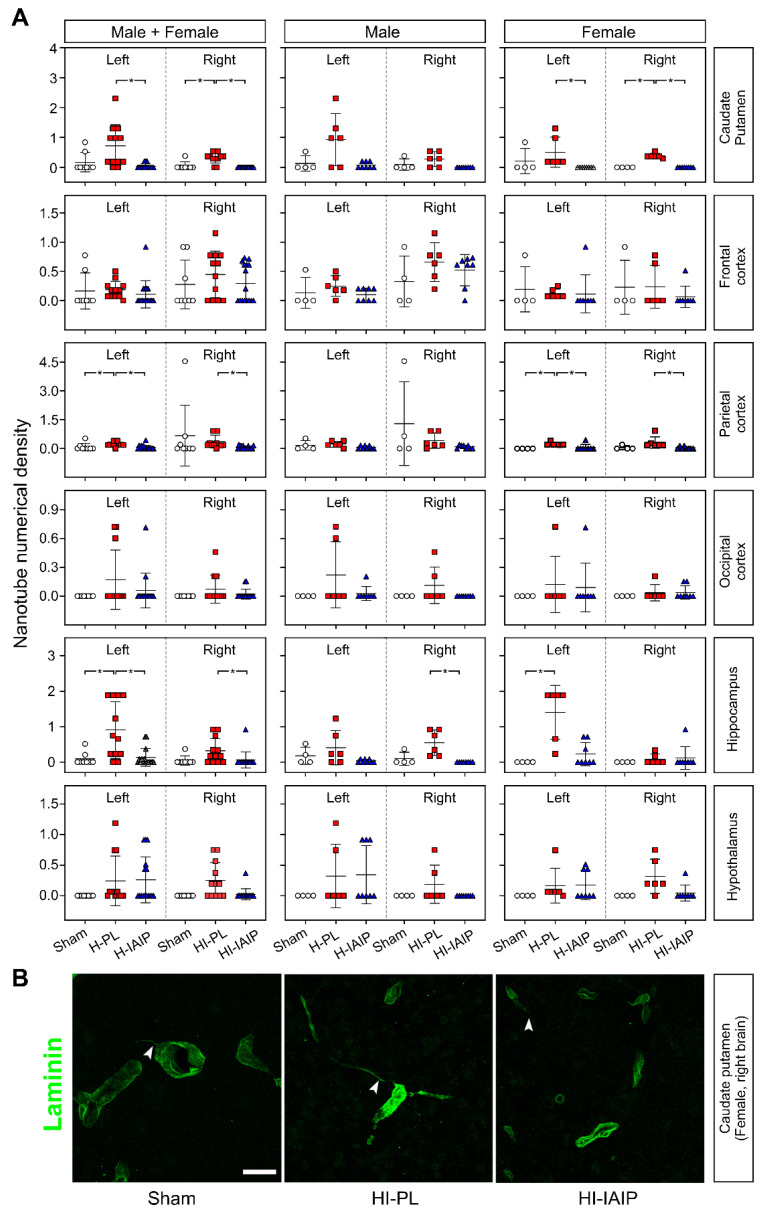
The effects of systemic treatment with IAIPs on TNT numerical density in neonatal rats exposed to systemic hypoxia (H-left) and hypoxia–ischemia (HI-right) injury. (**A**) Quantification of cumulative TNT linear density in the selected brain regions of total males + females and separately for the male and female left and right hemispheres. Data are mean ± SD. Male: sham = 4, H-PL/HI-PL = 6, H-IAIP/HI-IAIP = 8. Female: sham = 4, H-PL/HI-PL = 6, H-IAIP/HI-IAIP = 8. * *p* ≤ 0.05 for indicated comparisons. (**B**) Representative images of HI-right caudate–putamen, immunostained for laminin, showing the presence of TNTs connecting two microvessels (arrowhead, HI-PL) in neonatal female rats. In sham-operated and HI-IAIP-treated female rats, TNTs, although present (arrowhead), are rare. Scale bar: 25 µm.

**Figure 5 ijms-24-06743-f005:**
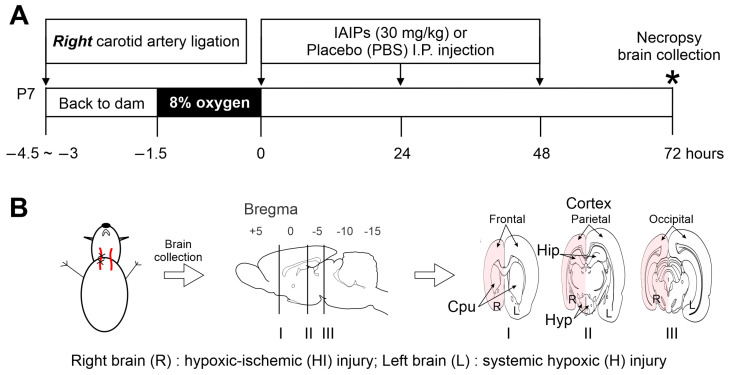
(**A**) Schematic of the study design. The pups were returned to the dams for 1.5 to 3 h after right common carotid artery ligation. Thereafter, the pups were exposed to 8% oxygen with balanced nitrogen for 90 min at a constant temperature of 36 °C. Thirty mg/kg of human blood-derived IAIPs (hIAIPs) or an equal volume of placebo (PBS) was given intraperitoneally (i.p.) immediately (0), 24, and 48 h after termination of hypoxia. Necropsy was performed on P10, and the brain was collected for immunohistochemical (IHC) staining (asterisk). (**B**) Schematic of the brain collection at P10.

**Table 1 ijms-24-06743-t001:** Schematic summary of statistically significant/non-significant differences in the quantification of the chosen morphometric parameters analyzed by laminin immunostaining in hIAIP- and placebo (PL)-treated neonatal rats. ↓ indicates reduced values, whereas ↔ indicates no statistically significant differences. Left: H-left hemisphere. Right: HI-right hemisphere.

Analysis	Brain Regions		HI-IAIP Vs. Placebo		
			Male + Female	Male	Female
Laminin area percentage	Caudate Putamen	Left	↓	↓	↓
		Right	↓	↔	↓
	Frontal cortex	Left	↓	↔	↓
		Right	↓	↔	
	Parietal cortex	Left	↓	↓	↔
		Right	↔	↓	↔
	Occipital cortex	Left	↓	↔	↔
		Right	↔	↔	↔
	Hippocampus	Left	↓	↓	↓
		Right	↓	↓	↔
	Hypothalamus	Left	↓	↓	↓
		Right	↓	↔	↔
Cumulative vessel length ×100/area (µm^2^)	Caudate Putamen	Left	↓	↔	↓
		Right	↔	↔	↔
	Frontal cortex	Left	↔	↔	↔
		Right	↔	↔	↔
	Parietal cortex	Left	↓	↔	↔
		Right	↔	↔	↔
	Occipital cortex	Left	↔	↔	↔
		Right	↔	↔	↔
	Hippocampus	Left	↔	↔	↔
		Right	↔	↔	↔
	Hypothalamus	Left	↔	↔	↔
		Right	↔	↔	↔
Nanotube linear density	Caudate Putamen	Left	↓	↔	↓
		Right	↓	↔	↓
	Frontal cortex	Left	↓	↔	↔
		Right	↔	↔	↔
	Parietal cortex	Left	↓	↔	↓
		Right	↓	↔	↓
	Occipital cortex	Left	↔	↔	↔
		Right	↔	↔	↔
	Hippocampus	Left	↓	↔	↔
		Right	↓	↓	↔
	Hypothalamus	Left	↔	↔	↔
		Right	v	↔	↔
Nanotube Numerical density	Caudate Putamen	Left	↓	↔	↓
		Right	↓	↔	↓
	Frontal cortex	Left	↔	↔	↔
		Right	↔	↔	↔
	Parietal cortex	Left	↓	↔	↓
		Right	↓	↔	↓
	Occipital cortex	Left	↔	↔	↔
		Right	↔	↔	↔
	Hippocampus	Left	↓	↔	↔
		Right	↓	↓	↔
	Hypothalamus	Left	↔	↔	↔
		Right	↔	↔	↔

## Data Availability

Further information regarding the resources, reagents, and data availability should be directed to the corresponding author and will be considered upon request.

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
