# Peer review of "Inter-Alpha Inhibitor Proteins Modify the Microvasculature after Exposure to Hypoxia–Ischemia and Hypoxia in Neonatal Rats"

_ijms, 2023, doi:10.3390/ijms24076743_

Round 1

Reviewer 1 Report

Manuscript details:

Journal: International Journal of Molecular Sciences

Manuscript ID: ijms-2264985

Title: Inter-alpha Inhibitor Proteins Modify the Microvasculature after 

Exposure to Hypoxia-Ischemia and Hypoxia in Neonatal Rats

Authors: Francesco Girolamo, Yow-Pin Lim, Daniela Virgintino, Barbara S. 

Stone street, Xiaodi F. Chen * Submitted to section: Molecular Neurobiology

In the current manuscript authors present morphometric analysis of vasculature upon hypoxia and hypoxia ischaemia insult in post natal rats, an experimental model representing hypoxia ischemia related damage in neonates, where there are currently limited therapeutic options. The authors present data on inter alpha inhibiting proteins (IAIP), which have been recently reported to reduce neuronal damage and inflammation along with LPS mediated BBB disruption. In the current manuscript the authors show that these IAIPs attenuate angiogenic parameters resulting from hypoxia and hypoxic ischemic damage in postnatal mice. The authors analysed a few parameters for vascular basement membrane coverage, vessel density, activation of angiogenesis and show that these parameters are increased in the H/HI model indicating angiogenic response that is potentially pathogenic. Post systemic administration of IAIPs, the authors show a rescue of these parameters closer to the sham operated healthy controls. There are however some concerns that need to addressed with respect to the critical role of IAIPs in cerebral angiogenesis.

Major:

The authors show all their data by immunohistochemistry and morphometric analysis, which is a major concern. The key aspects of IAIPs on angiogenic phenotypes should also be demonstrated at least in vitro using brain endothelial cells with and with IAIP treatment post H/HI insult. Also as BBB function is critical in these early phases post H/HI damage, the authors should investigate the impact of IAIPs on BBB by either fibrinogen, IgG or albumin staining post IHC (eg. PMID 35752654). Interestingly IAIPs bind hyaluronic acid, which has been shown to affect BBB function (PMID 29589805).

Minor:

There are several minor issues that need to be addressed that include typos throughout the text. The authors need to discuss the results and conclusions with respect to what the parameters stand for. This is particularly important in abstract and conclusion sections where laminin data should be discussed with respect to vascular basement coverage area and vessel density and not just area and length of the vessels.

Author Response

Response to the Reviewer 1’s Critiques:

We would like to thank reviewer 1 for his/her helpful suggestions, which have been incorporated into our revised manuscript. We very much appreciate the overall favorable comments and have addressed the concerns of reviewer 1 below and within the text of the manuscript.  Responses to specific comments are outlined below.

Reviewer 1:

  1. Major: 
  • The authors show all their data by immunohistochemistry and morphometric analysis, which is a major concern. The key aspects of IAIPs on angiogenic phenotypes should also be demonstrated at least in vitro using brain endothelial cells with and with IAIP treatment post H/HI insult.

Response:  We agree with the Reviewer that this would be important information to obtain.  As of yet, we have not generated such data and do not have residual fresh frozen tissue from the current study.  The invitro studies would be of great interest but are out of the scope of the present study. However, we have added this consideration to the Discussion (Page 15, Lines 514-519).

  • Also as BBB function is critical in these early phases post H/HI damage, the authors should investigate the impact of IAIPs on BBB by either fibrinogen, IgG or albumin staining post IHC (eg. PMID 35752654).

Response: Certainly, it would be of great interest to investigate whether treatment with hIAIPs affects BBB function in neonates after exposure to H/HI related insults. Treatment with hIAIPs could potentially decrease the concentration of molecules in the systemic circulation known to disrupt BBB integrity [1, 2]. We have previously shown that intraperitoneal administration of hIAIPs protect against LPS-induced BBB disruption in adult male mice [3], and may preserve BBB function through mechanisms associated with systemic cytokine suppression during inflammatory conditions [2]. Numerous papers suggest that IAIPs can suppress inflammation and cytokine production in a variety of animal models [4, 5].  In addition, we have previously shown that anti-cytokine antibodies against IL-1ß and IL-6 attenuate ischemia related disruption of the BBB in fetal sheep [6-8]. Therefore, it remains likely that IAIPs could also potentially attenuate H/HI related BBB disruption in neonatal rats [9]. The BBB permeability studies that we have performed in fetal sheep after ischemia and in adult mice after LPS exposure are complex and would require additional separate groups of animals, which is beyond the scope of the current manuscript. Although there are no accurate histopathological measures to determine kinetic BBB function [10-12], fibrinogen, IgG, and/or albumin have previously been used to estimate whether or not the BBB is intact [13]. However, these measures could be fraught with difficulties, as it is challenging to distinguish between the binding of fibrinogen, IgG, and/or albumin to the microvasculature versus actual penetration across the BBB into the brain parenchyma. Consistent with this view, others have suggested that post- mortem brain tissue, cannot be used to prove or disprove dysfunction of the BBB in the pathological tissues [14]. These concepts have been added to the discussion (Page 14, Lines 490-499).  Nonetheless, the impact of hIAIPs on BBB function after H/HI related brain injury in neonates is important and will be considered in future studies.

  • Interestingly IAIPs bind hyaluronic acid, which has been shown to affect BBB function (PMID 29589805).

 Response:  We appreciate this reviewer’s suggestion. We have modified the text and added the reference into the Discussion (Page 15, Lines 499-502).

  1. Minor: 
  • There are several minor issues that need to be addressed that include typos throughout the text.

Response:  We have made every effort to correct the typographical errors throughout the text. 

  • The authors need to discuss the results and conclusions with respect to what the parameters stand for. This is particularly important in abstract and conclusion sections where laminin data should be discussed with respect to vascular basement coverage area and vessel density and not just area and length of the vessels.

Response:  We have added to the abstract the following: “Our findings suggest that treatment with hIAIPs after exposure to H and HI in neonatal rats affects the laminin content of the vessel basal lamina and angiogenic responses in sex-related fashion.” (Page 1. Lines 32-34)

We have also amended our conclusions to emphasize that our findings suggest that treatment with hIAIPs after exposure to hypoxia and hypoxia-ischemia hIAIPs can attenuate the hypoxia and hypoxia-ischemia-related remodeling/thickening of the of the vessel basal lamina in neonatal rats. This effect is evidenced by the reductions to values that are similar to the laminin content values under normoxic conditions in the control neonatal rats [15] (Page 15, Lines 539-543).  This response could potentially be further investigated by examining pericyte-specific laminin isoforms [16, 17] (Page 12, Lines 390-392).

References:

  1. Disdier, C.; Stonestreet, B. S., Hypoxic-ischemic-related cerebrovascular changes and potential therapeutic strategies in the neonatal brain. J Neurosci Res 2020, 98, (7), 1468-1484.
  2. Koehn, L. M.; Chen, X.; Logsdon, A. F.; Lim, Y. P.; Stonestreet, B. S., Novel Neuroprotective Agents to Treat Neonatal Hypoxic-Ischemic Encephalopathy: Inter-Alpha Inhibitor Proteins. Int J Mol Sci 2020, 21, (23).
  3. Logsdon, A. F.; Erickson, M. A.; Chen, X.; Qiu, J.; Lim, Y. P.; Stonestreet, B. S.; Banks, W. A., Inter-alpha inhibitor proteins attenuate lipopolysaccharide-induced blood-brain barrier disruption and downregulate circulating interleukin 6 in mice. J Cereb Blood Flow Metab 2020, 40, (5), 1090-1102.
  4. Singh, K.; Zhang, L. X.; Bendelja, K.; Heath, R.; Murphy, S.; Sharma, S.; Padbury, J. F.; Lim, Y. P., Inter-alpha inhibitor protein administration improves survival from neonatal sepsis in mice. Pediatric research 2010, 68, (3), 242-7.
  5. Opal, S. M.; Lim, Y. P.; Siryaporn, E.; Moldawer, L. L.; Pribble, J. P.; Palardy, J. E.; Souza, S., Longitudinal studies of inter-alpha inhibitor proteins in severely septic patients: a potential clinical marker and mediator of severe sepsis. Crit Care Med 2007, 35, (2), 387-92.
  6. Sadowska, G. B.; Chen, X.; Zhang, J.; Lim, Y. P.; Cummings, E. E.; Makeyev, O.; Besio, W. G.; Gaitanis, J.; Padbury, J. F.; Banks, W. A.; Stonestreet, B. S., Interleukin-1beta transfer across the blood-brain barrier in the ovine fetus. Journal of cerebral blood flow and metabolism : official journal of the International Society of Cerebral Blood Flow and Metabolism 2015.
  7. Chen, X.; Sadowska, G. B.; Zhang, J.; Kim, J. E.; Cummings, E. E.; Bodge, C. A.; Lim, Y. P.; Makeyev, O.; Besio, W. G.; Gaitanis, J.; Threlkeld, S. W.; Banks, W. A.; Stonestreet, B. S., Neutralizing anti-interleukin-1beta antibodies modulate fetal blood-brain barrier function after ischemia. Neurobiol Dis 2015, 73, 118-29.
  8. Zhang, J.; Sadowska, G. B.; Chen, X.; Park, S. Y.; Kim, J. E.; Bodge, C. A.; Cummings, E.; Lim, Y. P.; Makeyev, O.; Besio, W. G.; Gaitanis, J.; Banks, W. A.; Stonestreet, B. S., Anti-IL-6 neutralizing antibody modulates blood-brain barrier function in the ovine fetus. FASEB J 2015, 29, (5), 1739-53.
  9. Mallard, E. C.; Williams, C. E.; Gunn, A. J.; Gunning, M. I.; Gluckman, P. D., Frequent episodes of brief ischemia sensitize the fetal sheep brain to neuronal loss and induce striatal injury. Pediatr Res 1993, 33, (1), 61-5.
  10. Saunders, N. R.; Dziegielewska, K. M.; Mollgard, K.; Habgood, M. D., Markers for blood-brain barrier integrity: how appropriate is Evans blue in the twenty-first century and what are the alternatives? Front Neurosci 2015, 9, 385.
  11. Saunders, N. R.; Dreifuss, J. J.; Dziegielewska, K. M.; Johansson, P. A.; Habgood, M. D.; Mollgard, K.; Bauer, H. C., The rights and wrongs of blood-brain barrier permeability studies: a walk through 100 years of history. Front Neurosci 2014, 8, 404.
  12. Saunders, N. R.; Liddelow, S. A.; Dziegielewska, K. M., Barrier mechanisms in the developing brain. Front Pharmacol 2012, 3, 46.
  13. Spitzer, D.; Guerit, S.; Puetz, T.; Khel, M. I.; Armbrust, M.; Dunst, M.; Macas, J.; Zinke, J.; Devraj, G.; Jia, X.; Croll, F.; Sommer, K.; Filipski, K.; Freiman, T. M.; Looso, M.; Gunther, S.; Di Tacchio, M.; Plate, K. H.; Reiss, Y.; Liebner, S.; Harter, P. N.; Devraj, K., Profiling the neurovascular unit unveils detrimental effects of osteopontin on the blood-brain barrier in acute ischemic stroke. Acta Neuropathol 2022, 144, (2), 305-337.
  14. Erickson, M. A.; Banks, W. A., Blood-brain barrier dysfunction as a cause and consequence of Alzheimer's disease. J Cereb Blood Flow Metab 2013, 33, (10), 1500-13.
  15. Gonul, E.; Duz, B.; Kahraman, S.; Kayali, H.; Kubar, A.; Timurkaynak, E., Early pericyte response to brain hypoxia in cats: an ultrastructural study. Microvasc Res 2002, 64, (1), 116-9.
  16. Gautam, J.; Xu, L.; Nirwane, A.; Nguyen, B.; Yao, Y., Loss of mural cell-derived laminin aggravates hemorrhagic brain injury. J Neuroinflammation 2020, 17, (1), 103.
  17. Gautam, J.; Zhang, X.; Yao, Y., The role of pericytic laminin in blood brain barrier integrity maintenance. Sci Rep 2016, 6, 36450.

Please see the revised manuscript with Changes and supplementary Figures.

Reviewer 2 Report

Girolamo et al. wanted to look at the effects of inter-alpha inhibitor proteins (IAIPs) on the small blood vessels in the brains of newborns who had been exposed to hypoxia-ischemia (HI). This work might help us learn more about how newborns' microvasculature develops in different ways. would like to thank the authors for their skilled creation of the work, which has captivating flow for the readership. However, have few concerns regarding the manuscript.

Line 18: Please give appropriate space around "treatmentor."   1. How was the dosage of human IAIPs (hIAIPs, 30 mg/kg) determined? Please give a reference for any dose-dependent studies that have already been done and explain why 30 mg/kg was chosen.

2. Please show a difference in the number of blood vessels using Rat Endothelial Cell Antigen-1 (RECA-1) or Nissl staining for neuronal damage.   3. Do you have information on the expression of claudin-5 in the microvasculature (as representative of tight junction proteins) in the different groups? If yes, please submit that.

4. Vessels that have been immunostained for the extracellular matrix protein Collagen IV or isolectin B4 are a strong and reliable way to find microvascular structures. This information will be a useful addition to the current data for figuring out how much the microvasculature changes after birth.

Author Response

Response to the Reviewer 2:

We would like to thank reviewer 2 for his/her helpful suggestions, which have been incorporated into our revised manuscript. We have addressed the concerns of reviewer 2 below and within the text of the manuscript.  Responses to specific comments are outlined below.

Reviewer 2:

  1. Line 18: Please give appropriate space around "treatmentor."

Response: The text has been corrected.

  1. How was the dosage of human IAIPs (hIAIPs, 30 mg/kg) determined? Please give a reference for any dose-dependent studies that have already been done and explain why 30 mg/kg was chosen.

Response: The 30 mg/kg dose of hIAIPs was selected based upon our previous studies that have shown that this dose of hIAIPs ameliorated pathological brain injury, infarct volume, and neuroinflammation in neonatal rats 3 days after exposure to moderate HI [1, 2]. Furthermore, we have shown that larger doses of hIAIPs (60 mg/kg and 90 mg/kg) did not provide additional benefits over the 30 mg/kg dose for behavior tasks or reductions in infarct volumes in neonatal rats after exposure to severe HI for 2 h [3]. We have modified the text in the Introduction (Page 2, Lines 87-92).

  1. Please show a difference in the number of blood vessels using Rat Endothelial Cell Antigen-1 (RECA-1) or Nissl staining for neuronal damage.

Response:  In our previous publication [1], we have shown Nissl cresyl violet staining for neuronal damage in neonatal rats using the same hypoxic-ischemic and hIAIP treatment procedures as in the current study. We have added this information to the Methods (Page 17, Lines 616-620).  To avoid self-plagiarism, we included a different Nissl-stained image to show neuronal damage as Supplementary Figure 1 and have referenced the original manuscript.

The reviewer raises a very good point with regards to the measurement of Rat Endothelial Cell Antigen-1 (RECA-1). Please see response # 5 below.

  1. Do you have information on the expression of claudin-5 in the microvasculature (as a representative of tight junction proteins) in the different groups? If yes, please submit that.

Response: We agree that it would be optimal to have measures of tight junction proteins (e.g. claudin-5) in the microvasculature.  However, we have not generated such data at this time.  However, this would be of great interest for future studies and we have added this consideration to the Discussion (Page 13, Lines 423-427; Page 14, Lines 502-504).

Moreover, although we have previously reported the time course of changes in some elements of the NVU in neonatal rats at 6 and 48 h after exposure to H and HI [4], we have not examined changes in the elements of the NVU 72 h after exposure to H/HI in neonatal rats or after exposure to a neuroprotective agent such as IAIPs, which we have shown has extensive neuroprotective properties after HI (1-3).

  1. Vessels that have been immunostained for the extracellular matrix protein Collagen IV or isolectin B4 are a strong and reliable way to find microvascular structures. This information will be a useful addition to the current data for figuring out how much the microvasculature changes after birth.

Response: Measuring Rat Endothelial Cell Antigen-1 (RECA-1) would certainly be of interest. However, the quantitative measures in the current study took over a year to complete because of the number of animals in each experimental group/controls and of the number of the selected CNS areas of interest. For this reason, and consistent with the original aims of our study, we used a pan-laminin antibody. In our experience, this method [4-6] is able not only to reveal endothelium-associated morphology of the microvessels but also blood-brain barrier/angiogenic morpho-functional features of the microvessels. In fact, laminin is a molecular component of the neurovascular unit (NVU) that is functionally associated with brain microvessel performance in both physiological and pathological conditions [4, 7]. Therefore, our analysis could be useful to suggest novel avenues for future in-depth studies regarding molecular changes in the NVU in response to exposure to H/HI and the potential for hIAIPs to protect the NVU/BBB. Moreover, laminin facilitates very early features of H/HI-dependent and pericyte-dependent changes, along with evidence of increased neo-collateralization. The latter, characterizes additional findings regarding the effects of H/HI-related changes on the initiation of angiogenesis by endothelial sprouting (Page 12,  Lines 372-377).

Future experiments will certainly be considered to focus on the examination of other interesting aspects of potential changes in the microvessels and other aspects of the NVU after exposure of neonatal rats to H/HI. It would be of great interest to define further the internal (endothelial) and external (parenchymal) basal laminae within the microvessels after exposure to H/HI and neuroprotective agents [8] (Page 15, Lines 514-519). In addition, it would be important to examine further other elements such as a more extensive examination of changes in tight junction proteins after exposure to H/HI and treatment with neuroprotective agents [4] (Page 14, Lines 502-504).

References:

  1. Chen, X.; Nakada, S.; Donahue, J. E.; Chen, R. H.; Tucker, R.; Qiu, J.; Lim, Y. P.; Stopa, E. G.; Stonestreet, B. S., Neuroprotective effects of inter-alpha inhibitor proteins after hypoxic-ischemic brain injury in neonatal rats. Exp Neurol 2019, 317, 244-259.
  2. Barrios-Anderson, A.; Chen, X.; Nakada, S.; Chen, R.; Lim, Y. P.; Stonestreet, B. S., Inter-alpha Inhibitor Proteins Modulate Neuroinflammatory Biomarkers After Hypoxia-Ischemia in Neonatal Rats. J Neuropathol Exp Neurol 2019.
  3. Koehn, L. M.; Nguyen, K.; Chen, X.; Santoso, A.; Tucker, R.; Lim, Y. P.; Stonestreet, B. S., Effects of Three Different Doses of Inter-Alpha Inhibitor Proteins on Severe Hypoxia-Ischemia-Related Brain Injury in Neonatal Rats. Int J Mol Sci 2022, 23, (21).
  4. Hatayama, K.; Riddick, S.; Awa, F.; Chen, X.; Virgintino, D.; Stonestreet, B. S., Time Course of Changes in the Neurovascular Unit after Hypoxic-Ischemic Injury in Neonatal Rats. Int J Mol Sci 2022, 23, (8).
  5. Girolamo, F.; Errede, M.; Longo, G.; Annese, T.; Alias, C.; Ferrara, G.; Morando, S.; Trojano, M.; Kerlero de Rosbo, N.; Uccelli, A.; Virgintino, D., Defining the role of NG2-expressing cells in experimental models of multiple sclerosis. A biofunctional analysis of the neurovascular unit in wild type and NG2 null mice. PLoS One 2019, 14, (3), e0213508.
  6. Virgintino, D.; Maiorano, E.; Errede, M.; Vimercati, A.; Greco, P.; Selvaggi, L.; Roncali, L.; Bertossi, M., Astroglia-microvessel relationship in the developing human telencephalon. Int J Dev Biol 1998, 42, (8), 1165-8.
  7. Zapata-Acevedo, J. F.; Garcia-Perez, V.; Cabezas-Perez, R.; Losada-Barragan, M.; Vargas-Sanchez, K.; Gonzalez-Reyes, R. E., Laminin as a Biomarker of Blood-Brain Barrier Disruption under Neuroinflammation: A Systematic Review. Int J Mol Sci 2022, 23, (12).
  8. Menezes, K.; Nascimento, M. A.; Goncalves, J. P.; Cruz, A. S.; Lopes, D. V.; Curzio, B.; Bonamino, M.; de Menezes, J. R.; Borojevic, R.; Rossi, M. I.; Coelho-Sampaio, T., Human mesenchymal cells from adipose tissue deposit laminin and promote regeneration of injured spinal cord in rats. PLoS One 2014, 9, (5), e96020.

Please see the revised manuscript with Changes and supplementary Figures.

Round 2

Reviewer 1 Report

Dear Authors,

I appreciate your efforts for understanding concern in regards your current study and I am satisfied with new revised version of Manuscript.  It is great study  definitely need followup work to find great alternative to current treatment.

Best regards

Reviewer 2 Report

The authors addressed the comments scientifically well and improved the manuscript.